# Evaluation of the Cross-Sectional Shape of μ–Grooves Produced in Stainless Steel 304 by Laser-Induced Etching Technique

**DOI:** 10.3390/mi12020144

**Published:** 2021-01-30

**Authors:** Jonghun Kim, Kwang H. Oh

**Affiliations:** Laser Advanced System Industrialization Center, Jeonnam Technopark Stiftung, Jeonnam 57248, Korea; kjh0099@jntp.or.kr

**Keywords:** laser micromachining, laser-induced etching, microgroove, cross-sectional shape, ray tracing

## Abstract

The variation in cross-sectional profile of a microgroove fabricated with focused and diverging laser irradiation is demonstrated with ray tracing. To verify the result of ray tracing, stainless-steel 304 microgrooves were manufactured utilizing the conventional lens-based and optical fiber-based laser-induced etching techniques in phosphoric acid solution. Three distinctive groove geometries, i.e., a flat surface with no groove, an intermediate stage groove, and a fully developed V-groove, were rendered for numerical analysis. For focusing mode, the first and second reflections were caused by high laser intensity and the second reflected beam could lead to variation in the groove shape such as a U-shaped groove or a V-shaped groove in accordance with etchant concentration. On the contrary, a weak laser entirely distributed at the groove sidewall could not induce a chemical reaction, leading to a V-shaped groove. The effect of process variables such as laser power (intensity) and etchant concentration on the cross-sectional profiles of a groove are closely examined through the computed simulation results.

## 1. Introduction

A microgroove is a key configuration of numerous microelectromechanical systems based on microthermal and microfluidic devices such as a micro heat pipe [1,2,3], micro pump [4,5], micro mixer [6,7], micro reactor [8,9], and micro fuel cell [10]. Due to the relative easiness in fabrication and simplicity in geometry, microgrooves are increasingly adopted in the design of microdevices; accordingly, researches about effective micromachining techniques of grooves are also increasing. Since the performance of microdevices is almost exclusively decided by the capillary pressure force and sidewall roughness, the controllable shape of a microgroove is absolutely demanded in fabrication technology. Among the fabrication technologies of a microgroove, laser precision microfabrication has been frequently adopted for the manufacture of microgrooves on a variety of materials in the past few decades. The laser material processing in association with a direct writing technique enables to easily provide a rapid prototype of microstructures with no mask or additional processes. Laser-based micromachining of metals has often been achieved with high-intensity pulsed laser irradiation. However, a large heat-affected zone (HAZ) with redeposition of evaporated or resolidificated material and mechanical load on the processed area can have a negative effect on the functional properties of microstructures [11,12].

As a possible approach to realize material removal while avoiding such issues, laser-induced etching (LIE) may be employed. In LIE, the workpiece is immersed in a liquid etchant that helps the fabrication of microgrooves with almost no melting or debris due to the cooling effect. The micromachining of a workpiece in LIE is done through a thermochemical reaction between the etchant and the workpiece, while the laser beam irradiates the workpiece to locally raise the surface temperature. A relatively simple setup compared to a picosecond or sub-picosecond pulsed laser machining system is required for LIE at a substantially lower equipment or maintenance cost. Due to these advantages of LIE, the fabrication of metallic microstructures using the LIE technique has been studied for various applications [13,14,15]. In addition, the cross-sectional profile of the microgroove can be adjusted between a rectangular shape and triangular shape by appropriately controlling the process variables such as the laser power and etchant concentration [16].

Despite several advantages, the typical lens-based LIE system still demands optical components such as lenses, mirrors, and a focusing lens. As a result, a complicated and routine optical alignment is frequently required, and the eye or skin of an operator could be exposed to a reflected or scattered laser beam. Furthermore, a relatively low productivity owing to the use of a single objective lens is not attractive for its application to manufacturing industries. Alternatively, an optical fiber as a low-loss light wave guide can be directly applied to the LIE system. By substituting a single optical fiber for the optical components, the whole system setup can be substantially simplified, while the safety trouble is naturally reduced. The laser beam is delivered through an optical fiber and, instead of using an objective lens, the fiber tip is directly brought to the surface of the workpiece to irradiate and micromachine the workpiece. In our previous studies, we reported that LIE utilizing an optical fiber as the laser beam guide could be effectively applied to the fabrication of metallic microgrooves [17], and we demonstrated the manufacture of grooves with only a triangular cross-sectional profile regardless of process variables.

In this study, to examine the effects of laser irradiation modes on the shape of a microgroove, ray tracing was conducted as an optical rendering. Ray tracing is a well-established technique for generating an image by tracing the path of light originating from an object. The effects of a focused laser beam from an objective lens and a diverging laser beam from an optical fiber on the cross-sectional profiles of the microgrooves are investigated using ray tracing software LightTools^®^ (Optical Research Associates, Pasadena, CA, USA).

## 2. Materials and Methods

### 2.1. System Configuration

Figure 1a shows the schematic diagram of a conventional lens-based LIE system. A diode-pumped solid-state continuous wave laser with a maximum power of 8 W and a center wavelength of 532 nm was used as the heat source. The laser beam was expanded five times and then tightly focused using a focusing lens (5×, focal length = 40 μm, working distance = 37.5 mm, numerical aperture = 0.14, depth of focus = 14 μm) on the surface of the workpiece after passing through a linear polarizer and quarter-wave plate. The laser beam was circularly polarized at the surface of the workpiece to avoid a possible polarization dependence of the etching process [18]. The direct writing of grooves was done by translating the workpiece using a high-resolution (1 μm/pulse) motorized *XYZ* stage. The workpiece was placed inside a Teflon^®^ chamber that was installed on the *XYZ* stage.

In conventional LIE, the laser light from a source travels through several optical elements such as a beam expander, a polarizing beam splitter, and quarter-wave plate before being focused by a focusing lens and irradiating the workpiece as shown in Figure 1a. A disadvantage of this configuration is that alignment of the laser beam through the optical elements is frequently required, which is tedious and potentially dangerous to a user. To overcome the shortcomings of conventional LIE, an optical fiber can be employed for laser beam delivery, as illustrated in Figure 1b. A pulsed Nd:YAG laser (center wavelength = 532 nm, maximum power = 60 W, pulse duration = 150 ns, repetition rate = 10 KHz) was used to manufacture microgrooves. The laser beam was launched into the optical fiber by a focusing lens (5×, focal length = 40 μm, numerical aperture = 0.14, depth of focus = 14 μm). Commercially available multimode fibers with core diameter of 50 μm and 105 μm (cladding diameter = 125 μm, numerical aperture = 0.22 for both fibers) were used as the light waveguide and machining tool. Note that, unlike a conventional LIE system in which the laser beam is transported to the workpiece through many optical components, the optical fiber delivers the laser beam directly to the surface of the workpiece with no intermediate optical elements, and no focusing lens is used to focus the laser beam on the workpiece. The fiber tip was housed inside a titanium jig with a jig mount and installed on a high-resolution (1 μm/pulse) motorized *XYZ* stage. The workpiece was placed inside a Teflon^®^ chamber through which the etchant is continuously circulated by a peristaltic pump.

A stainless-steel 304 foil (Fe72/Cr18/Ni10) with a thickness of 500 μm was used as the workpiece. Additionally, phosphoric acid (H_3_PO_4_, 85%) diluted with distilled water was used as the etchant to etch the stainless-steel 304 foil. All etching processes were monitored in real time with a charge-coupled device (CCD) camera. Etching results were examined using a scanning electron microscope after an ultrasonic cleaning of the workpiece.

### 2.2. Modeling for Ray Tracing

Figure 2 shows an illustration of the geometry of the model for ray tracing. The material used for modeling was stainless steel 304. To investigate the variation in laser intensity over the groove surface for the abovementioned two different types of laser irradiation, a three-dimensional ray tracing was performed with cartesian coordinates.

The ray tracing calculation was conducted for three different groove geometries as presented in Table 1: (i) a plane surface with no groove, (ii) a developing groove in an intermediate stage with width and depth of 30 and 85 μm, respectively, for the focused irradiation condition and 100 and 85 μm for the diverging beam from an optical fiber, and (iii) a completed V-shaped groove structure with width and depth of 50 and 300 μm, respectively, for the focused irradiation and 100 and 300 μm for the diverging laser beam.

The distances between the optical source and the workpiece in the simulation were 40 mm for the focused beam condition and 150 μm for the diverging beam condition. For an examination of laser intensity distribution on groove walls, optical receivers for detecting the incident rays were installed at the top, sidewall, and bottom of the groove models.

Ray tracing is a rendering technique based on the reflection and refraction of light. Therefore, to obtain the distribution of laser intensity over the irradiated surfaces, the reflectivity of the material should be known. A reflectivity value of 0.7 (at 532 nm) was used for the stainless steel 304 [19,20], and a specular reflection was assumed. The rays not reflected at the groove surface were allowed to transmit through the surface so that the transmitted rays could be detected by the receivers of the model.

## 3. Results and Discussion

### 3.1. Etch Threshold Temperature

To investigate the surface temperature variation of the workpiece during laser irradiation, the workpiece temperature was calculated using the thermal model in our previous research [21]. Temperature fluctuations of the workpiece during laser irradiation are a major concern in LIE, and the quality of the machined area depends significantly upon melting and solidification of the workpiece.

Prior to the etching experiments, the etch threshold laser power was investigated via irradiation of the diverging continuous wave (CW) laser beam. Figure 3 shows the hemisphere-like microstructure produced with an experimentally verified threshold laser power of about 0.7 W. The diameter and depth of the etched structure were about 60 μm and 2 μm, respectively. For this threshold power, the computed maximum temperature rise at the workpiece surface was about 360 K for a stationary laser beam; for a 2 μm/s scan speed in the CW etching experiment, the maximum dwelling time of the laser beam at a location was 40 s, as presented in Figure 4.

### 3.2. Comparison of Microgrooves Fabricated by Two Different Laser Irradiation Modes

The etching reaction and, thus, fabrication results are highly dependent upon the etchant concentration. Since the velocity of etchant solution varies with concentration, the etchant flow over the workpiece and, more importantly, inside a microstructure under fabrication can differ significantly for low- and high-concentration cases. Figure 5 shows an example of the stainless-steel 304 microgroove manufactured at two different H_3_PO_4_ (etchant) concentrations of 10% and 40% with a lens-based LIE system. The grooves fabricated at these concentrations showed a clear difference in the cross-sectional profile. Those fabricated at the low concentration in Figure 5a,b had a rather uniform width over depth with a round U-shaped bottom. On the other hand, the grooves for the high concentration in Figure 5c,d showed a triangular cross-section due to a gradual decease in groove width with increasing depth. Note that the surface pattern of the grooves for both 10% and 40% cases show no difference. The observed difference in cross-sectional profile was considered to be due to the viscosity changes between these two concentrations. The viscosity of 40% (≈3.8) H_3_PO_4_ is about three times greater than that of the 10% (≈1.3) etchant [22,23], which may significantly the influence on the etchant flow inside a narrow groove. For the thermochemical reaction to be sustained, fresh etchant should be continuously supplied into the groove. At high concentrations, however, viscous resistance may hinder an easy refreshment of etchant into the groove, possibly leading to the loss of radicals or even bleaching of etchant. A decease in radicals would result in a natural decrease in groove width because, in this case, the etching reaction would be confined to only near the center of the Gaussian beam where the maximum intensity occurs.

For the optical fiber-based LIE, grooves with only a triangular cross-section were fabricated at all concentrations. The reason for this result is that the laser beam emerging from a fiber tip diverges very rapidly, where the degree of divergence depends upon the numerical aperture of an optical fiber. Due to the divergence, the average irradiance decreases rapidly with the distance from the fiber tip. Accordingly, as the etching process is completed and, thus, the groove surface retreats, the etching reaction is active only in the middle of the beam at which the irradiance remains over a threshold value. In the optical fiber-based LIE case, the aspect ratio of the groove can be adjusted by varying etchant concentration and the number of scans over the same groove. Figure 6 shows a representative example of groove fabrication for which a highly uniform and clear surface pattern with no undesired thermal effects was observed. The cross-sectional profile of this groove had a uniform triangular shape and was reproducible. These results demonstrate that the proposed optical fiber-based LIE is an effective method to fabricate metallic microgrooves.

The width of microgrooves that can be fabricated with the optical fiber-based LIE depends on the diameter of optical fiber, and about minimum 50 μm at the surface can be achieved. Note that, for the conventional lens-based LIE, the groove width can be as small as 15 μm because, in this case, the laser beam spot can be focused to a much smaller size. The depth of grooves varies with laser power but it can also be adjusted by varying the number of scans. For a single scan, the groove depth reaches 100 to 200 μm depending on laser power; however, by repeatedly scanning over the same groove, the depth can be increased to 300–400 μm.

### 3.3. Optical Rendering

For experiments using the lens-based case, the radius of laser spot at the surface of workpiece was 7 μm as measured with a knife edge method for *P* = 2 W and *f* = 40 mm. On the contrary, the 1/*e*^2^-radius of the laser beam emitted for the optical fiber was about 75 μm for *P* = 2 W and *D_gap_* = 150 μm. Since the laser intensity is inversely proportional to the laser spot area, the temperature of the irradiated surface for these two different modes of LIE may also have a large difference. Furthermore, since multiple reflections of the laser beam at the groove walls can affect the groove wall temperature, the cross-sectional shape of a groove may also be related to the intensity of the reflected beam.

Figure 7 shows the computed intensity distributions at the nonetched surface of stainless steel 304 for the “focusing mode” and “diverging mode”. The number of rays irradiating the workpiece was limited to 3,000,000 due to calculation time. For the calculation, an optical receiver was numerically placed beneath the surface to detect the incident ray, and the sizes of optical receivers for the focusing mode and diverging mode were 50 μm × 50 μm and 200 μm × 200 μm, respectively. The computed beam diameters agreed with the value of experiments. For the focusing mode, the maximum intensity reached about 3165 W/mm^2^, while that for the diverging mode was only about 1.7 W/mm^2^. For these laser intensities, the maximum surface temperatures calculated using the thermal model for the focusing and diverging modes were about 1490 K and 470 K, respectively, higher than the etch threshold temperature of 360 K. Therefore, etching of the flat workpiece was expected for both irradiation modes, subsequently developing into a groove. During this initial period, the groove shapes were expected to be similar, although their sizes could be different.

However, once a groove is formed, the irradiation pattern over the groove walls and the associated development of groove shape are expected to be quite different between the two modes. Figure 8 shows the intensity distribution over the sidewalls of the intermediate stage groove (width = 30 μm and depth = 85 μm) for the focusing and diverging modes. For the focusing mode, absorption of the incident rays over the sidewall increased almost exponentially with groove depth with little absorption at the upper part of the sidewall (Figure 8a). The absence of rays near the top of the groove was due to the difference in propagation directions of rays and the angle of sidewalls. In other words, since the incident beam was confined within a narrow angle (less than about 16° by the numerical aperture (NA) of the objective lens, NA = 0.14) that was smaller than the angle formed by the groove walls (about 20° for the intermediate stage groove of consideration), most of the rays directly hit the lower portion of the groove wall, as shown in Figure 8b. Accordingly, the groove width at the surface did not change noticeably, whereas the bottom portion of the groove was continuously etched away, forming a deeper groove.

On the contrary, the diverging mode showed an entirely different tendency. Unlike the focusing mode, relatively low intensity with a maximum value of about 1.3 W/mm^2^ was distributed almost over the entire sidewall of the intermediate stage groove (width = 100 μm and depth = 85 μm), as shown in Figure 8c. This was possibly due to a characteristic of the diverging beam delivered through the optical fiber. In other words, since the size of the incident beam was larger than the groove width at the surface, the rays could irradiate from the top to the bottom of the sidewall, as shown in Figure 8d.

The intensity distributions at the bottom of intermediate-stage grooves for the focusing and diverging modes are presented in Figure 9. As expected, in the case of focusing mode, Figure 9a, most of the incident rays directly propagated to the bottom of the groove, possibly inducing a strong etching reaction near the bottom portion and leading to the formation of a high-aspect-ratio groove.

For the diverging mode, a very weak concentration of incident rays at the center of the groove bottom occurred. Moreover, the laser beam intensity at the bottom of the groove (see Figure 9c) was almost the same as that at the sidewall of Figure 8c. Therefore, it was expected that the sidewall and bottom could be etched away at a similar rate, maintaining the original triangular cross-sectional profile.

Figure 10a,b present the intensity distributions at the sidewall of a fully developed microgroove with the focusing mode. The modeled width and depth of the microgrooves were 50 μm and 300 μm, respectively, and the calculated groove angle was about 10°.

The intensity distribution over the sidewall of the fully developed groove in Figure 10a showed two distinctive peaks, namely, the “first reflection” and “second reflection” peaks, both being sufficiently high to induce the etching reaction. Between these two reflection peaks, the more critical one in the determination of groove shape could be the second reflection peak. As demonstrated earlier in Section 3.2, thanks to the low viscosity at the relatively low etchant concentration of 10%, the removal of etch byproduct and supply of fresh etchant can occur relatively easily. Therefore, the thermochemical reaction by the first and second reflection peaks can be induced at the sidewall near the bottom of the groove, allowing the fabrication of a U-shaped groove for the low etchant concentration of 10% if a focused laser beam is used. For the case of the relatively higher etchant concentration of 40%, microbubbles and etch byproducts cannot easily escape from the groove due to the increased viscosity and, in this case, the laser beam may be scattered by the bubbles or insoluble byproducts, losing power before the second reflection; accordingly, etching of the sidewall near the bottom portion cannot occur, which ultimately results in the formation of a groove.

For the diverging mode, the laser intensity over the sidewall (see Figure 10c) and at the bottom of the groove (see Figure 11b) for the fully developed groove appeared too low to induce thermochemical reaction. To verify this, a numerical calculation of the surface temperature variation was carried out with the maximum laser intensity on the sidewall (0.6 W/mm^2^), and the results are presented in Figure 12. For this intensity, the computed maximum temperature rise at the sidewall was about 349 K, which is far less than the etch threshold temperature of 360 K for a stationary laser beam. Therefore, etching of the sidewall and bottom of the groove could not occur any further for the fully developed groove. According to these results for ray tracing, it could be verified that the fabrication of both U-shaped and V-shaped grooves was feasible for the focusing mode by appropriately adjusting the etchant concentration, whereas only a V-groove could be formed in the diverging mode regardless of the etchant concentration.

## 4. Conclusions

The influence of laser irradiation modes such as focused laser beam and diverging laser beam on the cross-sectional shape of a microgroove was investigated using ray tracing for three groove geometries, i.e., an initial flat surface with no groove, a developing groove of intermediate depth, and a fully developed V-groove.

Regardless of the laser irradiation modes, etching of the workpiece is expected in case of the flat surface with no groove due to the initial high laser intensity. However, once the groove is formed, an obvious difference appears between the focusing mode and the diverging mode and, thus, the probable cross-sectional profiles of a groove are noticeably distinguished. For focusing mode related to the typical lens-based LIE, the high first and second reflected laser intensities could sufficiently induce a thermochemical reaction. Above all, the second reflected laser intensity has a deeper relationship with the cross-sectional profile and, according to the etchant concentration, a U-shaped groove or a V-shaped groove can be produced. On the other hand, for diverging mode related to the optical fiber LIE, since the laser intensity distributed on the sidewall and bottom portion is too weak, no further etching occurs and the V-groove is maintained. According to this study, it was found that the ray tracing results enable prediction of the shape of the microgroove that can be manufactured through LIE.

It is believed that the LIE could be a practical micromachining technique for the manufacture of microthermal and microfluidic devices such as a micro heat pipe, micro heat spreader, and micro reactor, based on metallic microgrooves.

## Figures and Tables

**Figure 1 micromachines-12-00144-f001:**
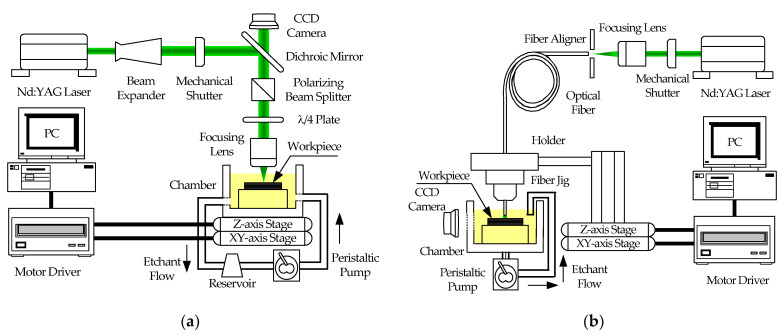
Schematic diagram of the (**a**) conventional lens-based LIE setup and (**b**) optical fiber-based LIE setup.

**Figure 2 micromachines-12-00144-f002:**
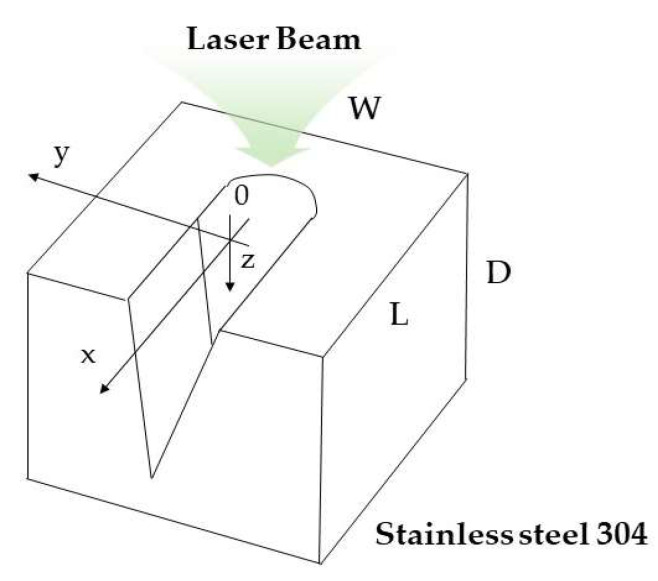
Schematic diagram for the physical model of stainless-steel 304 workpiece for optical analysis.

**Figure 3 micromachines-12-00144-f003:**
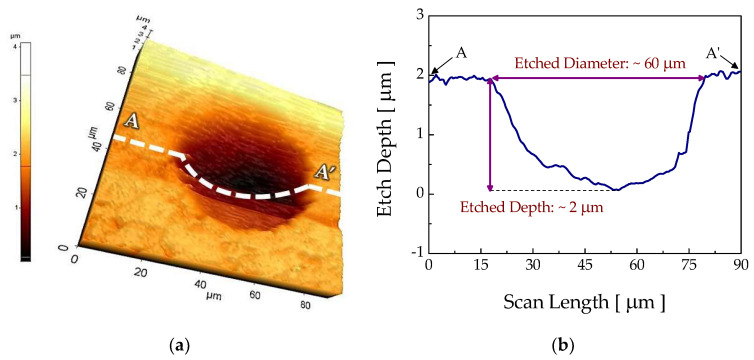
(**a**) Atomic force micrograph for the hemisphere-like structure processed with the continuous wave (CW) laser irradiation lasting 40 s and (**b**) the cross-section of A–A′. Other process variables were *P* = 0.7 W and *D_core_* = 105 μm.

**Figure 4 micromachines-12-00144-f004:**
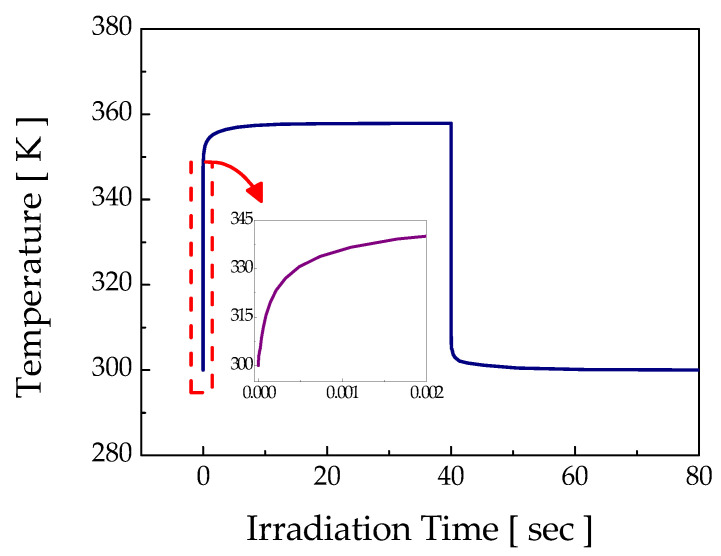
Workpiece surface temperature variation at the threshold etch condition.

**Figure 5 micromachines-12-00144-f005:**
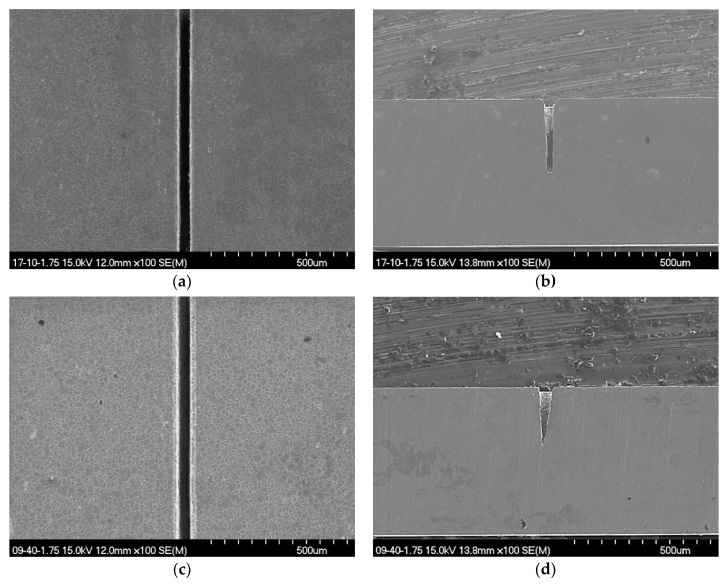
Scanning electron micrographs for the surface morphologies and cross-sectional profiles of stainless-steel 304 microgrooves processed at two different H_3_PO_4_ concentrations of 10% (**a**,**b**) and 40% (**c**,**d**) utilizing the lens-based LIE. Other process variables were a laser power of 1.75 W and scan speed of 10 μm/s.

**Figure 6 micromachines-12-00144-f006:**
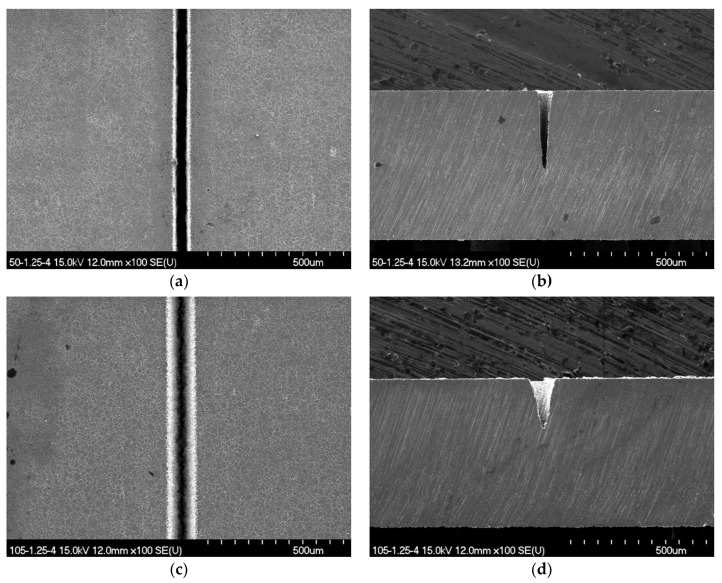
Scanning electron micrographs for the surface morphologies and cross-sectional profiles of stainless-steel 304 microgrooves fabricated in 10% H_3_PO_4_ with two different core diameters using the optical fiber-based LIE: (**a**,**b**) 50 μm core fiber and (**c**,**d**) 105 μm core fiber. Other process variables were a laser power of 1.25 W and scan speed of 4 μm/s.

**Figure 7 micromachines-12-00144-f007:**
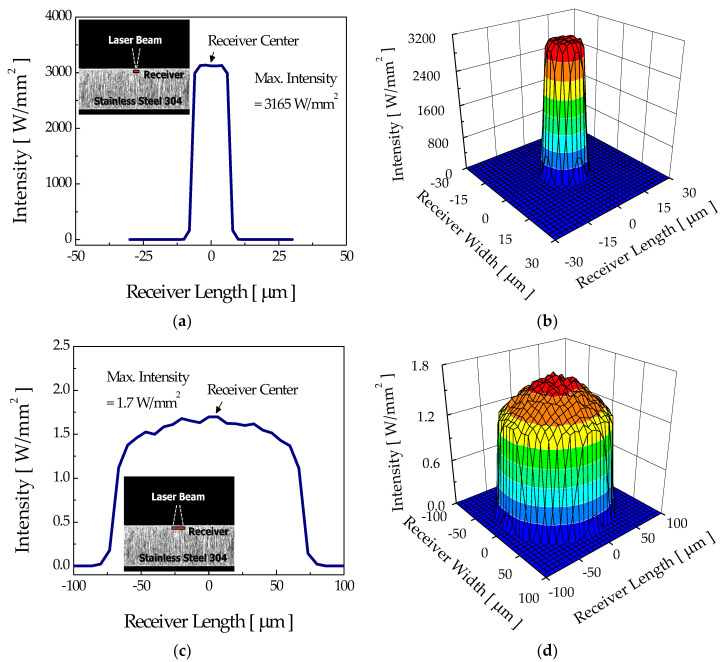
Surface intensity distributions and three-dimensional profiles of the flat surface with no groove which appeared in the (**a**,**b**) focusing mode and (**c**,**d**) diverging mode.

**Figure 8 micromachines-12-00144-f008:**
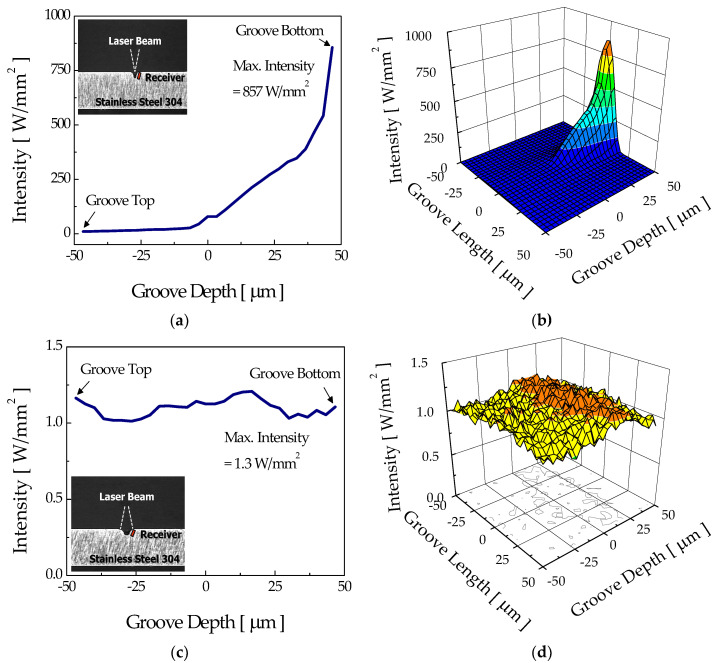
Sidewall intensity distributions and three-dimensional profiles of the intermediate stage groove which appeared in the (**a**,**b**) focusing mode and (**c**,**d**) diverging mode.

**Figure 9 micromachines-12-00144-f009:**
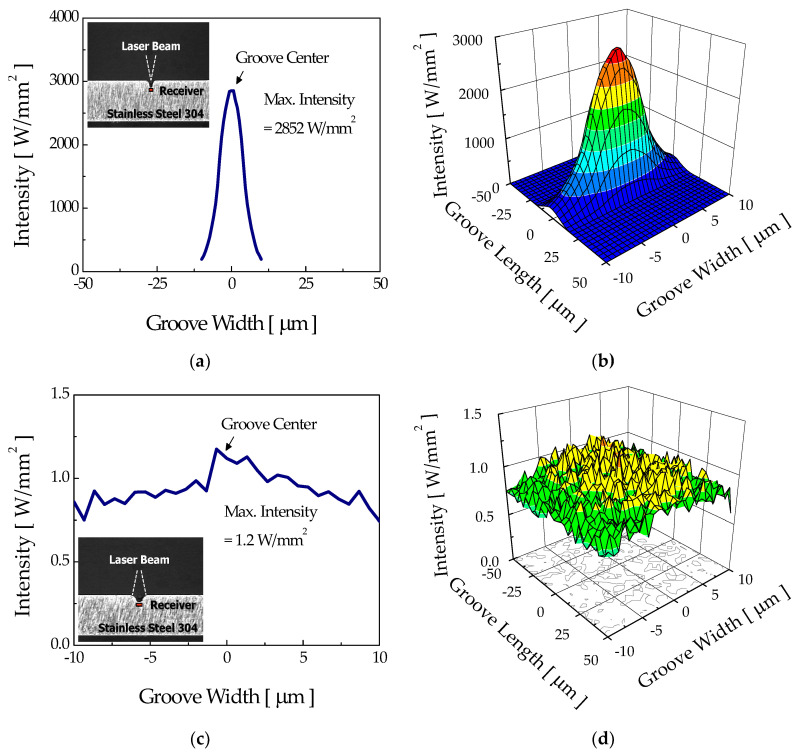
Bottom intensity distributions and three-dimensional profiles of the intermediate stage groove which appeared in the (**a**,**b**) focusing mode and (**c**,**d**) diverging mode.

**Figure 10 micromachines-12-00144-f010:**
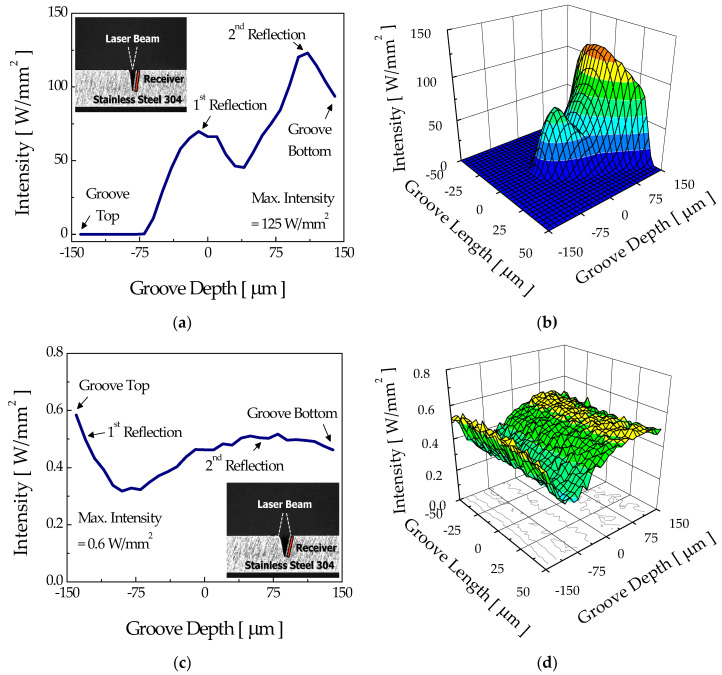
Sidewall intensity distributions of the fully developed V-groove which appeared in the (**a**,**b**) focusing mode and (**c**,**d**) diverging mode.

**Figure 11 micromachines-12-00144-f011:**
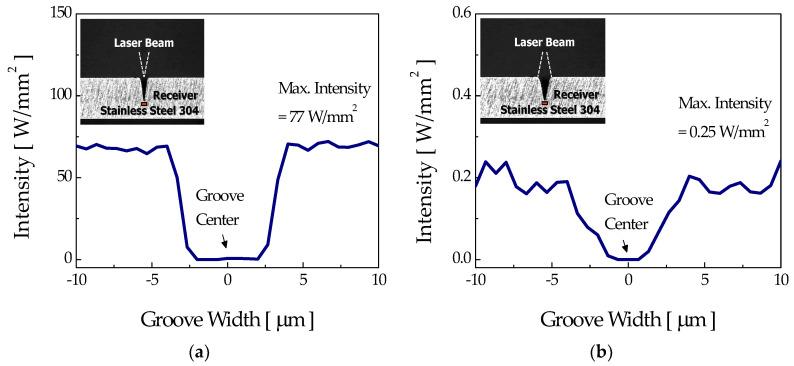
Bottom intensity distributions of the fully developed V-groove which appeared in the (**a**) focusing mode and (**b**) diverging mode.

**Figure 12 micromachines-12-00144-f012:**
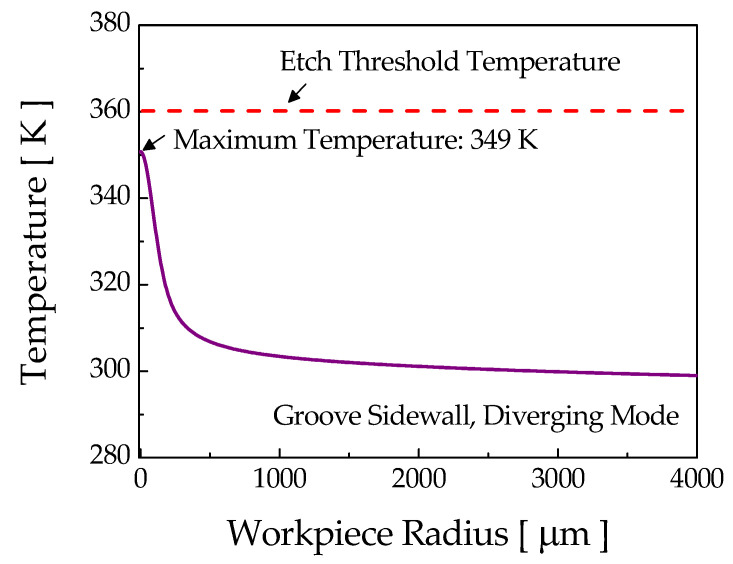
Sidewall temperature variation of the microgroove for the diverging mode.

**Table 1 micromachines-12-00144-t001:** Three types of groove geometry for ray tracing calculation.

Groove Geometry		Irradiation Mode	
		Focused Beam	Diverging Beam
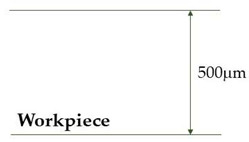 (**Flat surface with no groove**)	Width	0	0
	Depth	0	0
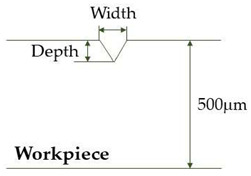 (**Intermediate stage groove**)	Width	30 μm	100 μm
	Depth	85 μm	85 μm
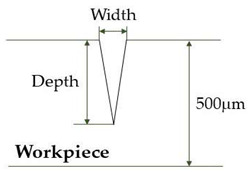 (**Fully developed V-groove**)	Width	50 μm	100 μm
	Depth	300 μm	300 μm

## Data Availability

Data available in a publicly accessible repository.

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
