# Peer review of "Evaluation of the Cross-Sectional Shape of μ–Grooves Produced in Stainless Steel 304 by Laser-Induced Etching Technique"

_micromachines, 2021, doi:10.3390/mi12020144_

Round 1
Reviewer 1 Report
This paper claim that focused laser beam is useful for laser-induced etching (LIE). But it is too obvious fact for laser process engineers. In other word, the comparison between focused beam and diverging beam has less impact to the laser society. However, as my knowledge, this is the first study that tries to reveal the underneath thermochemical phenomena. Recently, industry starts to have an interest to LIE process. And many studies have focused on process itself. But this study focuses on the thermochemical phenomena. Thus, I want to give the authors a credit for this attempt. There are several questions and comments for better paper.
- For the lens-based LIE, how the laser beam is delivered? Mirrors and lens? Or Fiber and lens?
- Also, please describe your lens information. (Focal length or NA)
- Why don’t you use collimator or focusing lens at the end of fiber tip? Is there any reason?
- What is the NA of Fiber?
- What if the intensity of diverging laser beam is matched with focused laser beam? Do you think that those case brings the same result? (Please briefly mentioned with proper logics, you don’t need to perform additional experiment or simulation)
- Please check your grammar.
- Ex. page1. Line 14: were cause -> were caused…
Other than those comments, the result is interesting and helpful for peer researchers.
Author Response
Q1. For the lens-based LIE, how the laser beam is delivered? Mirrors and lens? Or Fiber and lens?
A1. We described in detail regarding experiments for the conventionl lens-based LIE and optical fiber-based LIE in the revised paper.
Q2. Also, please describe your lens information. (Focal length or NA)
A2. As mentioned in the revised paper, the focusing lens has a focal length of 14um and an numerical aperture of 0.14.
Q3.Why don’t you use collimator or focusing lens at the end of fiber tip? Is there any reason?
A3.Some experimets with lensed fiber were conducted. The cross-sectional shape of a groove was manufactured in a fan-shape similar to a triangular shape. However, many experiments could not be conducted due to frequent damage and breakage of the fiber tip.
Q4. What is the NA of Fiber?
A4. NA of optical fiber(multimode fiber) used in experiments was 0.22 and it was also expressed in the revised paper.
Q5. What if the intensity of diverging laser beam is matched with focused laser beam? Do you think that those case brings the same result? (Please briefly mentioned with proper logics, you don’t need to perform additional experiment or simulation)
A5. In laser-induced etching, the laser light is used as a heat source that induces a chemical reaction between the workpiece and the etchant, rather than directly processing the sample. Therefore, it is thought that the average power should be considered rather than the laser intensity relatthe beam size. We tried to increase the average power several times, but the experiments could not proceed smoothly due to frequent fiber damage.
Q6 & Q7. Please check your grammar.
A6 & A7. The pointed out were revised and a grammar review was conducted for the entire revised paper.
Reviewer 2 Report
The paper presents results on the shape of microgrooves manufactured with laser‐induced etching using a ray-tracing analysis. In my opinion the paper is interesting, but it may be improved. A schematic diagram of the experimental setups (lens and fiber-based) should be included, as well as a description of the characteristics of the lasers used for the fabrication of the microgrooves shown in figures 2, 4 and 5 (similarly to what was done in reference 17). There is only one sentence in the conclusions that mention: “It was confirmed that the ray tracing results were almost identical to the shape of the microgrooves manufactured by the LIE experiment”. A detailed comparison between experimental results and results obtained ray tracing should be made. Finally, the writing language should be significantly improved, including the title of the paper (for example, something like “Evaluation of the cross-sectional shape of microgrooves produced in stainless steel by laser‐induced acid etching” seems more appropriate).
Author Response
Q1. A schematic diagram of the experimental setups (lens and fiber-based) should be included, as well as a description of the characteristics of the lasers used for the fabrication of the microgrooves shown in figures 2, 4 and 5.
A1. A description of the experiments (schematics, method, etc.) was written in the revised paper and also the laser properties were explained at the same time.
Q2. The writing language should be significantly improved, including the title of the paper.
A2. Tilte of the revised paper was changed to the recommended tilte ("Evaluation of the Cross-sectional Shape of m-grooves Produced in Stainless Steel 304 by Laser-induced Etching Technique")
Round 2
Reviewer 2 Report
The paper has been improved but a comparison between the experimental results shown in figures 5 and 6 and results obtained ray tracing is still missing.
Line 268: where it is “groove shape are expected quite different between” it should be “groove shape are expected to be quite different between”.
Line 399: where it is “using by” it should be only “by” or “using”.
Author Response
Q.1 The paper has been improved but a comparison between the experimental results shown in figures 5 and 6 and results obtained ray tracing is still missing.
A.1. The purpose of ray rat tracing in this study is to predict the shape of microgrooves that can be fabricated using the conventional LIE and optical fiber-based LIE. Especially, in the conventional LIE, it has been inferred that U-shaped or V-shaped grooves can be manufactured depending on the concentratoin of the etchant because the laser intensity of the bottom portion maintains a sufficient temperature to induce the etching. Therefore, since the results for Figs. 5 and 6 are not compared with the ray tracing results, the sentence of the conclusion has been modifiled as follows. "According to this study, it was found that the ray tracing results enables prediction of the shape of the microgroove that can be manufactured through LIE."
Q.2 Grammar check
A.2 Line 268 and 399 have been revised.